# Optimization of a Three-Dimensional Culturing Method for Assessing the Impact of Cisplatin on Notch Signaling in Head and Neck Squamous Cell Carcinoma (HNSCC)

**DOI:** 10.3390/cancers15225320

**Published:** 2023-11-07

**Authors:** Alinda Anameriç, Arkadiusz Czerwonka, Matthias Nees

**Affiliations:** Department of Biochemistry and Molecular Biology, Medical University of Lublin, 20-093 Lublin, Poland; 62005@student.umlub.pl (A.A.); arkadiusz.czerwonka@umlub.pl (A.C.)

**Keywords:** head and neck squamous cell carcinoma (HNSCC), 3D culture, 3D coculture, Notch signaling, patient-derived fibroblasts (CAFs), tumor microenvironment (TME), extracellular matrix (ECM), cisplatin resistance, chemosensitivity, gamma-secretase inhibitors, crenigacestat

## Abstract

**Simple Summary:**

Most experimental research on head and neck squamous cell carcinoma (HNSCC) relies on two- or three-dimensional (2D/3D) cell- and tissue-culture model systems. Commonly used methods like 2D monolayer cultures or 3D organoid models typically lack critical components of the tumor microenvironment (TME), such as cancer-associated fibroblast (CAFs), that likely influence chemosensitivity versus drug resistance of the tumor cells. In addition, based on the small experimental scale, it is often difficult to isolate a sufficient number of cells, proteins, or RNA from miniaturized 3D cultures for subsequent molecular analyses. In this manuscript, we describe a novel, more robust, and simultaneously larger-scale model system in which tumor cells and CAFs spontaneously generate tumor microtissues that mimic the architecture and histology of HNSCC biopsies. We have used this “3D sheet model” to investigate the assumed functional connection between NOTCH signaling and sensitivity to the chemotherapeutic drug cisplatin in HNSCC.

**Abstract:**

Head and neck squamous cell carcinoma (HNSCC) is a prevalent cancer type, with cisplatin being a primary treatment approach. However, drug resistance and therapy failure pose a significant challenge, affecting nearly 50% of patients over time. This research had two aims: (1) to optimize a 3D cell-culture method for assessing the interplay between tumor cells and cancer-associated fibroblasts (CAFs) in vitro; and (2) to study how cisplatin impacts the Notch pathway, particularly considering the role of CAFs. Using our optimized “3D sheet model” approach, we tested two HNSCC cell lines with different cisplatin sensitivities and moderate, non-mutated NOTCH1 and -3 expressions. Combining cisplatin with a γ-secretase inhibitor (crenigacestat) increased sensitivity and induced cell death in the less sensitive cell line, while cisplatin alone was more effective in the moderately sensitive line and sensitivity decreased with the Notch inhibitor. Cisplatin boosted the expression of core Notch signaling proteins in 3D monocultures of both lines, which was counteracted by crenigacestat. In contrast, the presence of patient-derived CAFs mitigated effects and protected both cell lines from cisplatin toxicity. Elevated NOTCH1 and NOTCH3 protein levels were consistently correlated with reduced cisplatin sensitivity and increased cell survival. Additionally, the Notch ligand JAG2 had additional, protective effects reducing cell death from cisplatin exposure. In summary, we observed an inverse relationship between NOTCH1 and NOTCH3 levels and cisplatin responsiveness, overall protective effects by CAFs, and a potential link between JAG2 expression with tumor cell survival.

## 1. Introduction

Head and neck squamous cell carcinomas (HNSCCs) are prevalent cancers, primarily affecting the oral cavity, nasopharynx, hypopharynx, and larynx [1,2]. Presently, HNSCC is the seventh most common cancer worldwide. The incidence of HNSCC is expected to increase by 30% by the year 2030. Furthermore, more than 50% of HNSCC patients face lung metastasis and recurrence less than 3 years after diagnosis [3]. Chronic exposure to tobacco smoke, excessive alcohol consumption, and infection with high-risk human papillomavirus strains (mainly HPV16 and 18) are the most common risk factors associated with the development of HNSCC [4].

Despite the increasing incidence, the therapeutic landscape of HNSCCs remains challenging. Standard treatments for primary cancers involve surgery and local irradiation. Advanced-stage, recurrent, and metastatic tumors (R/M HNSCCs) have only very limited treatment options and often develop resistance to the most commonly used drugs such as 5-fluorouracil (5-FU), taxanes (docetaxel and paclitaxel), and platinum-based drugs including cisplatin and carboplatin [5]. Cisplatin is also a fundamental component of the EXTREME therapeutic regimen (Erbitux in First-Line Treatment of Recurrent or Metastatic Head and Neck Cancer) that is now the established standard-of-care therapy for advanced R/M HNSCC, which combines the recombinant antibody cetuximab (Erbitux) with fluorouracil plus a platinum compound (cisplatin or carboplatin) [6]. However, only a fraction of patients respond for longer periods to these therapies, and improvements in disease-free or overall patient survival are limited, typically at a significant reduction in the quality of life. Newer targeted treatments, most notably addressing immune checkpoint inhibitors like PD-1 and PD-L1 (e.g., ipilimumab, nivolumab, or bevacizumab), show positive response rates in only 15–20% of patients, preferentially in “hot” tumors with overt immune infiltration.

Cisplatin, also known as cis-diamminedichloroplatinum II (CDDP), is a platinum-based drug widely employed for various cancer types, including HNSCC. The cellular uptake of cisplatin is facilitated by copper membrane transporters 1 and 2 (CTR1 and CTR2) in mammalian cells, including cancer cells. Cisplatin exposure results in excessive DNA damage, such as the accumulation of double-strand breaks, which blocks cell division and induces programmed cell death or apoptosis. Although cisplatin initially exhibits a response rate of up to 50% in therapy, the remaining half of patients do not demonstrate any significant response. Most patients eventually acquire increased resistance to cisplatin, resulting in cancer recurrence and highly aggressive local or distant metastases [7,8,9,10]. The putative molecular mechanisms leading to acquired cisplatin response are complex and reviewed in detail in [7], and include pathways promoting stemness and increased tumor cell plasticity, such as the Notch and the Wnt pathway.

Exploring the molecular changes promoting HNSCC progression, the Notch signaling pathway emerges as a significant player. Notch signaling plays a pivotal role in regulating cell differentiation, proliferation, and apoptosis across multicellular organisms, including mammals. It represents a central mechanism in cell fate determination and lineage-specific tissue differentiation or cell maturation, while Notch signaling is simultaneously associated with potentially promoting and maintaining stem cell characteristics in normal and tumor cells [7,8,9,10,11]. Mammals possess four distinct Notch receptors (NOTCH1–4), along with five Notch ligands: Jagged 1 and 2 (JAG1 and JAG2), and the delta-like ligands (DLL1, DLL3, and DLL4). Canonical and non-canonical activation of the Notch signaling pathway (including schematic representations) with a focus on head and neck cancer are represented in recent papers [11,12].

Notch signaling is also involved in the initiation and progression of certain cancer types, most prominently, in acute lymphoblastic leukemia (T-ALL), triple-negative breast cancers (TNBC), and HNSCC—but in very different or even opposing ways. Mutations in all four Notch receptors have been identified in HNSCC, with NOTCH1 mutations being the most prevalent (17–20%), followed by NOTCH2 (7–9%), NOTCH3 (5%), and NOTCH4 (2–3%). In contrast to T-ALL and TNBC, mutations in HNSCC are almost exclusively loss-of-function (LoF) mutations that result in truncated, non-functional receptors. Nevertheless, contradictory reports exist that suggest a correlation between increased expression of NOTCH receptors and progression, metastasis, and drug resistance in HNSCC. For example, it has recently been reported that overexpression of NOTCH1 in HNSCC may be associated with poor patient survival and therapy outcome [13], expanding on earlier reports that increased NOTCH1 expression may be associated with cisplatin resistance in HNSCC [14]. Another recent report further suggests the involvement of NOTCH3 (over-) expression in metastatic HNSCC [15]. On the other hand, Notch ligands are rarely mutated, deleted, or amplified, but JAG1 and JAG2 are overexpressed in more than 30% of HNSCC [16]. It is therefore possible that altered expression and local activation of Notch signaling by Notch ligands may actively contribute to tumor progression, metastasis, and acquired chemoresistance. A core question of this research relates to the exact localization and cell types in which Notch receptors and ligands are expressed and interact, thus possibly contributing to tumor progression and drug resistance. This points to the putative role of the tumor microenvironment in Notch signaling, and the possibility that cancer-associated fibroblasts, endothelial cells, or even immune cells may contribute to altered Notch signaling activity in HNSCC, compared to normal oral mucosa.

In recent years, the promising and potent targeted drug crenigacestat (CRE) has emerged as a potential treatment for HNSCC. Crenigacestat (LY3039478) is an oral Notch and a selective gamma-secretase inhibitor (GSI) with an IC50 of 0.41 nM [17]. By blocking gamma-secretase, CRE is currently in early-stage clinical trials against various solid cancers, and was reported to effectively disrupt the Notch signaling pathway [18], inhibit tumor growth, reduce fibrosis, and potentially overcome cisplatin resistance [19]. Here, we use CRE in combination with cisplatin to test the hypothesis that targeting both DNA damage response and Notch signaling may result in potentially synergistic effects. The functional and molecular relationship between Notch signaling and cisplatin resistance has been only poorly investigated thus far [20,21,22] and almost exclusively in routine two-dimensional monolayer cell cultures.

To test cellular behavior and drug responses, scientists frequently employ in vitro cell-culture methodologies and develop cell-based model systems. Cell-culture techniques are crucial in understanding drug responses in cancer research. Traditional two-dimensional (2D) monolayer cell cultures involve growing cells on flat plastic dishes. “Classic” two-dimensional or 2D monolayer cultures on plastic plates have several critical limitations for drug discovery, including their unnatural cell shape, and dependence on or even addiction to artificial cell adhesion, which leads to significant differences in gene and protein expression compared to in vivo tissues. Two-dimensional monolayer cultures also suffer from highly artificial drug delivery, pro-inflammatory hyperstimulation of growth and proliferation, and altered metabolization. In contrast, three-dimensional (3D) “organotypic” cultures rely on the growth of spontaneously forming organoids embedded within or on top of hydrogels or scaffolds that mimic the extracellular matrix (ECM) of tumors or normal tissues [23]. In contrast, tumor cell aggregates forming in non-adherent plates are typically called spheroids and lack strong cell–cell interaction, polarization, and differentiation features induced by the ECM. Three-dimensional cultures potentially offer a more accurate representation of tissue- and disease-specific drug effects and promise to mimic key aspects of chemosensitivity and drug resistance. They exhibit higher cell density, more intense cell–cell interactions, and the formation of tissue-like and often functional structures, and are often characterized by oxygen and metabolite gradients often including hypoxic regions [24]. Moreover, the activity of cell survival and anti-apoptotic pathways (such as integrin, PI3Kinase, and AKT pathways), combined with cell fate decision mechanisms (like Notch or WNT), can massively differ between 2D and 3D cultures, thus significantly affecting drug sensitivity in vitro. Additionally, the expression and activity of membrane transporters associated with multidrug resistance can vary between 2D and 3D cultures [25]. However, no “gold standards” or even minimum conditions and requirements for 3D cultures are widely accepted for performing in vitro chemosensitivity assays, and comparative studies are still largely missing. In previous studies, our laboratory has developed and described miniaturized 3D model systems that promote the growth of three-dimensional, multicellular structures such as organoids and or other types of cell aggregates (such as grape-like cell clusters) in the so-called “sandwich model” first described in [26]. This is based on a specialized well-in-a-well design that allows the generation of ECM layers without the characteristic meniscus formation, which typically represents a problem for confocal microscopy and high-content imaging of 3D organoids and other multicellular structures. The “sandwich” design further promotes the formation of a precisely defined layer of three-dimensional organoids or tissue-like, multicellular structures embedded between two sheets of extracellular matrix. This layered design also facilitates the use of differential matrices, or differential matrix density, in the bottom and top layers, which can be useful to promote cell motility and invasion towards a specific direction [27]. This design, which uses an industry-standard 96-well plate format, is ideal for automated imaging approaches, high throughput, and high content screening, and specifically facilitates confocal microscopy or light sheet microscopy. This, in turn, can be combined with automated morphometric image analysis methods [28], and allows simultaneous tracking of tumor cells and, for example, cancer-associated fibroblasts [29,30]. However, the focus of the “sandwich model” is on miniaturization and automation. This results in small tissue-like structures which do not readily allow further investigation which requires larger numbers of cells, tissues, organoids, or tissue-like aggregates, e.g., for subsequent molecular analysis of proteins, mRNA, or metabolites. We, therefore, aimed to develop a larger-scale model system that results in formation and easy access to a more substantial numbers of cells, multicellular 3D organoids, or complex microtissues, which spontaneously form by the combination of multiple cell types (such as tumor cells and fibroblasts).

Most 3D models still represent reductionist and over-simplified cell cultures lacking most of the cellular components of the tumor microenvironment (TME), such as the fibroblasts. It is very likely, however, that stromal elements—which are found in all solid tumors—will significantly affect drug sensitivity and the development of acquired drug resistance in cancer chemotherapy. For this reason, we used 3D cocultures of tumor cells and cancer-associated fibroblasts (both immortalized cell lines and primary CAFs isolated from patient tumors) to address this common experimental shortcoming and further hoped to spontaneously generate complex, three-dimensional, and tissue-like structures in our advanced 3D model system, which mimics the histology and architecture of HNSCC in patient tumors. We called this increased-volume experimental system the “3D sheet model”, in contrast to the established sandwich model. In the visual abstract and Figure 1, we outline the two main methods used for generating in vitro microtissues in this publication, described in detail in Section 2.

## 2. Materials and Methods

### 2.1. Cell Lines and Cell Culture

We have used the cell lines UT-SCC-24B and UT-SCC-42B, both originating from the laboratory of Reidar Grenman at the University of Turku, Finland, and kindly provided by Auria Biobank/Turku (Turku, Finland). These lines originate from recurrent or metastatic HNSCC, respectively, and were selected based on differential cisplatin sensitivity (as described in [31]. Both cell lines do not harbor any mutation in the 4 NOTCH receptors. Notch signaling appears fully active, including the expression of all four Notch proteins (full length and cleaved) and five ligands. Detailed information on genomic, genetic, and epigenetic changes or mRNA gene expression and drug sensitivity are available from [31]. Information on the origin and genetic characteristics of these cell lines is also available from the Cellosaurus depository:
UT-SSC-24B: https://www.cellosaurus.org/CVCL_7827 (accessed on 31 October 2023)UT-SCC-42B: https://www.cellosaurus.org/CVCL_7848 (accessed on 31 October 2023)

Cell lines were cultured in complete medium (DMEM with 10% FBS (Corning, New York, NY, USA), 1% penicillin/streptomycin (SIGMA-Aldrich, Darmstadt, Germany), and 1% glutamax (SIGMA-Aldrich) at 37 °C and 5% CO_2_ until the cells reached 90% confluence.

### 2.2. Use of the Sandwich Model in µ-Angiogenesis 96-Well Plates

Rat tail collagen type I from Corning (0.75 mg/mL) and Matrigel**^®^** basement membrane extract, also from Corning (2 mg/mL), were mixed with Dulbecco’s DMEM to create the bottom gel (10 μL per well) in a 96-well µ-angiogenesis plate (ibidi GmbH, Munich, Germany). The plate was centrifuged at 200× *g* for 10 min to ensure a flat gel surface formation. After incubation at 37 °C and 5% CO_2_ for 30 min, UT-SCC-24B and UT-SCC-42B cells were stained with Vybrant™ DiD cell-labeling solution from Invitrogen™ (V22887). hTERT-immortalized and GFP-labeled CAFs were used for cocultures [30]. For the upper gel, Matrigel (1 mg/mL) and collagen (0.375 mg/mL) were mixed with cell suspension and added on top of the bottom gel (20 μL per well). The µ-angiogenesis plate was centrifuged at 100× *g* for 20 min, and incubated at 37 °C and 5% CO_2_ for 5 days. Differences in 3D morphology between embedded and “on top” cell seeding are demonstrated in Appendix A.

### 2.3. Isolation of Patient-Derived Fibroblasts from Tumor Biopsies

Building upon the foundational knowledge from the sandwich model, we sought to further understand the tumor microenvironment by isolating patient-derived fibroblasts. Single-cell suspension of patient tumor samples was prepared according to a modified protocol using a 50 mL Falcon tube containing 25 mL DMEM (Corning, NY, USA) with 100 μg/mL Primocin (Invivogen, Toulouse, France) [32,33]. The tissue biopsy was minced into ~1 mm^3^ pieces and digested in 0.25% trypsin for 30–60 min. Undigested tissue fragments were removed using a 100 µm cell strainer (Corning) and a second 40 µm strainer was used for single-cell isolation. The filtered cells were centrifuged, washed to remove debris, and placed in Nunc™ EasYFlask™ Cell Culture Flasks (Thermo Fisher Scientific, Grand Island, NY, USA). After removing the supernatant, complete medium (DMEM supplemented with 10% FBS (Corning), 1% penicillin/streptomycin antibiotic solution (SIGMA), and 100 µg/mL Primocin antifungal solution) was added to the culture and cultured at 37 °C and 5% CO_2_ until the cells reached 70–90% confluence. After this time, the medium was changed to DMEM medium only with the addition of 10–18% FBS and Pen/Strep antibiotic solution. Any remaining tumor colonies were gradually removed by spontaneous differentiation, induced upon increasing FBS concentration stepwise (up to 18%). This method is described in [32,33] Patient-derived primary tissues were obtained with permission and according to the guidelines of the Institutional Review and Ethical Board of the Medical University of Lublin.

### 2.4. Preparation of “3D Sheet Model” and “3D Sandwich Model” in Costar^®^ 12-Well Plate

After the successful isolation of patient-derived fibroblasts, we aimed to optimize our 3D culture conditions. In the 3D sandwich model, we prepared a bottom gel by mixing Corning^®^ Collagen I rat tail (1 mg/mL) and Corning^®^ Matrigel^®^ Matrix (2 mg/mL) with Dulbecco’s DMEM to a final volume of 350 μL per well in a Costar^®^ 12-Well plate. After centrifuging at 200× *g* for 10 min, the plate was incubated for 45 min. The upper gel layer, consisting of Matrigel (1 mg/mL), collagen (0.375 mg/mL), and a cell suspension of 2.5 × 10^5^ cells, was then added on top. The setup was centrifuged at 100× *g* for 20 min and incubated at 37 °C in a 5% CO_2_ atmosphere for 5 days. For the 3D sheet model, we added a 350 μL solution of Corning^®^ Collagen I rat tail (1 mg/mL), Corning^®^ Matrigel^®^ Matrix (2 mg/mL), and Dulbecco’s DMEM to each well of a Costar^®^ 12-Well plate, and, after incubation at 37 °C for 1.5 h, the gel was polymerized. We then seeded 2.5 × 10^5^ tumor cells for monocultures and, for cocultures, we seeded 1.25 × 10^5^ tumor cells alongside 1.25 × 10^5^ patient-derived fibroblasts on the polymerized gel with 75 μL of medium, supplementing with an additional 100 μL of DMEM after 1 day. This small amount of medium allowed an active air–liquid interface (ALI) to form in the well. The cells were then cultured for 5 days. After the decision to choose the sheet model over the sandwich model for further analyses, firstly immunofluorescence and then qRT-PCR analysis were performed for chosen wells. After 5 days of untreated growth, 3D sheet cultures were exposed to drug combinations of 5 µM or 10 μM cisplatin (from a 1 mM PBS stock) with 5 μM crenigacestat (LY3039478) from a 10 mM stock in 100% DMSO. Following treatment, cultures were incubated for an additional 2 days at 37 °C in 5% CO_2_. Both the sandwich model and 3D sheet model were imaged by a Nikon ECLIPSE Ti confocal microscope (Melville, NY, USA) and EVOS M5000 Imaging System (Thermo Fisher).

### 2.5. Immunofluorescence Staining

Immunofluorescence (IF) staining was performed using a modified version of a previously published method [34]. The medium was removed from each well of the cell-culture plate, and the wells were washed three times with PBS without calcium and magnesium ions. The cells and cell aggregates in each well were fixed using a 2-step protocol: first with 4% paraformaldehyde in PBS for 15 min at room temperature and then with a chilled 3:1 methanol/acetone mix (chilled at −20 °C) for 5 min at −20 °C. The wells were subsequently washed three times with PBS. The wells were incubated in a cell penetration buffer for 30 min to enhance antibody penetration, containing 0.2% Triton X-100, 0.3 M glycine, and 10% DMSO (Sigma-Aldrich) dissolved in PBS [35]. After another wash with PBS, the wells were incubated with a blocking buffer containing 0.1% Triton X-100, 3% bovine albumin serum (BSA, Sigma-Aldrich), and PBS for 1 h at room temperature with gentle shaking. The primary antibodies (Anti-E-Cadherin from Abcam, Boston, MA, USA, #ab76055, and Anti-Vimentin antibodies from Abcam, #ab45939) were diluted 1:100 in the blocking buffer and incubated with the wells for 2 h at room temperature. Subsequently, the wells were washed three times with washing buffer (PBS Buffer 1X with 0.1% Tween 20) and incubated with secondary antibodies: Alexa Fluor™ 488 conjugated goat anti-Rabbit IgG (H&L) Cross-Adsorbed Antibody, and Alexa Fluor™ 555-conjugated goat anti-Mouse IgG (H&L) Cross-Adsorbed Antibody, both from Invitrogen, at a dilution of 1:200 for 1 h at room temperature. After three washes with blocking buffer, the wells were incubated with Hoechst 33342 solution (Chemodex, St. Gallen, Switzerland) at a 1:300 dilution in washing buffer for 15 min at room temperature. Finally, the wells were washed three times with a washing buffer for the last time. Throughout the IF staining process, the plate was kept strictly protected from light to prevent the bleaching of fluorophores. Three-dimensional microtissues were imaged using a Nikon confocal microscope and individual stack images were merged into 3D “image cubes” to highlight the three-dimensional structure and depth of the emerging microtissues.

### 2.6. Isolation of Cell Cultures from Matrigel Collagen Mixture Gel

After the WST-8 assay, wells were washed with 300 μL of phosphate-buffered saline (PBS) devoid of calcium and magnesium ions. For coated plates, 100 μL trypsin-EDTA (Gibco-Aldrich) (0.25%) was used to detach the cells by incubation at 37 °C in a humidified incubator for 10 min. For digestion of cells and tissue-like aggregates/structures in 3D sheet and sandwich models, a solution of 2.5 mg/mL of Dispase II (neutral protease, Worthington Biochemical, Lakewood, NJ, USA) was used to dissolve the collagen/Matrigel gel by incubating the plate at 37 °C in a humidified incubator for 30–45 min. Subsequently, the cell suspension was collected in 15 mL plastic centrifuge tubes and centrifuged at 4 °C for 10 min. After discarding the supernatant, the cell pellet was washed with PBS lacking calcium and magnesium ions, and centrifuged at 4 °C for 10 min.

### 2.7. Quantitative PCR (qRT-PCR) Analysis

To further investigate the molecular changes in our models with different CAF/tumor cell ratios, we performed quantitative, reverse-transcription PCR (qRT-PCR) analyses. Cells were harvested using Dispase II (Sigma-Aldrich D4693) and total RNA was extracted using the RNeasy Micro Kit (QIAGEN. Hilden, Germany). cDNA synthesis was carried out using the High-Capacity cDNA Reverse Transcription Kit (Applied Biosystems, Thermo Fisher). qRT-PCR analysis was conducted on a LightCycler 480 II instrument (Roche, Penzberg, Germany) using PowerUp SYBR Green Master Mix (Applied Biosystems, Thermo Fisher). Relative mRNA expression was calculated using the 2^−∆∆Ct^ method, normalized to GAPDH. Fold change (FC) values were then used to quantitate differences in mRNA expression levels. The significance of differential expression was analyzed using the GraphPad Prism 9 program (La Jolla, CA, USA). Primer sequences were obtained from the OriGene database or designed with Benchling software (San Francisco, CA, USA), based on the Ensembl Genomes database, and synthesized by GenoMed (Warsaw, Poland). All primers were tested for specificity and sensitivity.
GAPDH (forward 5′-GTGGAGTCTACTGGTGTCTTC-3′, reverse 3′-GTGCAGGAGGCATTGCTTACA-5′),NOTCH1 (forward 5′-CAACTGCCAGAACCTTGTGC-3′ reverse 3′-GGCAACGTCAACACCTTGTC-5′),NOTCH3 (forward 5′-GCAGATGGCTCAACGGCACTG-3′, reverse 3′-GGGGTCTCCTCCTTGCTATCCTG-5′).

Fold change values in the range of 0–0.75 was considered as downregulated (* *p*  ≤  0.05) and values 1.5–10 as upregulated (* *p*  ≤  0.05), respectively.

### 2.8. WST-8 Metabolic Assay

WST-8 assay was performed according to the protocol mentioned before by optimizing it for 12-well plates instead of 96-well plates [36]. WST-8 assay was originally developed as an alternative to MTT and other metabolic endpoint assays [37]. In contrast to widely used MTT or CellTitreBlue assays, WST-8 does not require cell lysis and absorption can be directly measured in the microtiter plate without transfer. After measuring metabolic activity in 2D or 3D culture by WST-8 assay, these can be further used for other purposes, such as protein or mRNA extraction. Briefly, WST-8 reagent solution (5 mM WST-8, 0.2 mM 1-methoxy PMS, and 150 mM NaCl) was prepared. The WST-8 reagent solution was diluted 1:10 with DMEM (40 μL WST-8 reagent to 400 μL DMEM to each well), added to the wells, and incubated at 37 °C for 2 h. After incubation, absorbance at 450 nm was measured using a multi-plate reader.

### 2.9. Protein Extraction and Western Blotting

Beyond these analyses, understanding protein dynamics was crucial, especially when exploring a pathway like Notch signaling which has very significant implications in tumor biology, especially in HNSCC. After cell isolation from gels, total proteins were isolated using radioimmunoprecipitation assay (RIPA) buffer containing 50 mM Tris-HCl, 150 mM NaCl, 1.0% NP-40, 0.5% sodium deoxycholate, and 0.1% sodium dodecyl sulfate, supplemented with protease inhibitor cocktails (Sigma-Aldrich). The cells were incubated in RIPA buffer on ice for 1 h, followed by centrifugation at 14,000 rpm for 20 min at 4 °C to remove undissolved debris. The protein solution supernatants were transferred to clean tubes and protein concentrations were determined using the Pierce™ BCA Protein Assay Kit [38] Protein samples (40 μg) and pre-stained protein markers (PageRuler™ Plus Prestained Protein Ladder and Spectra™ Multicolor Broad Range Protein Ladder, all from Thermo Scientific™) were denatured by heating in the presence of β-mercaptoethanol, and separated on modified 4–15% gradient SDS-PAGE gels. The gradient gels were prepared by combining solutions of 15% and 4% separating gel in a serological pipette. After polymerization of the separation gel, a 4% stacking gel solution was added. A 10-well 1.0 mm Mini-PROTEAN^®^ Comb (Bio-Rad, Hercules, CA, USA) was placed into the stacking gel before polymerization. After gel preparation, protein samples and pre-stained protein markers were separated by SDS-PAGE electrophoresis. The proteins were then transferred to a PVDF polyvinylidene difluoride membrane using a Bio-Rad Trans-Blot SD Semi-Dry Transfer Cell for 60 min. Ponceau S staining was performed on the membranes for total protein normalization [39]; membranes were blocked for 1 h at room temperature (RT) in blocking buffer (1X Tris-buffered saline (TBS) with 0.5% (*v*/*v*) TWEEN 20 and 5% (*w*/*v*) nonfat dried milk powder) and incubated overnight at 4 °C with the primary antibodies. The antibodies used in this study were NOTCH1 (D1E11) XP^®^ Rabbit mAb (1:1000), NOTCH3 (D11B8) Rabbit mAb (1:1000), JAG2 (C23D2) Rabbit mAb (1:1000), and Anti-rabbit IgG, HRP-linked Antibody #7074 (1:2000). After washing the membranes with TBS containing 0.1% Tween-20, they were incubated for 1 h at RT with the secondary antibody (Anti-rabbit IgG, HRP-linked Antibody #7074). The membranes were then washed three times for 10 min each with TBS/0.1% Tween-20. Protein detection was performed using Pierce™ ECL Western Blotting Substrate. Densitometry analysis was conducted for all blots by using the Fiji program, and values were normalized to Ponceau S staining for each condition. Also, beta-Actin staining was used as a loading control, but showed significant differences in expression due to varying cell-culture conditions and treatments; therefore, we decided not to use any housekeeping genes as loading controls and currently present Ponceau-S staining of blots after transfer as a reliable protein loading control instead. Membrane stripping was performed using a mild stripping protocol as described in reference [40].

### 2.10. Statistical Analysis

All experiments including WST-8 assay and RT-PCR were performed at least in triplicate. Statistical analyses of all samples were performed using GraphPad Prism 8.0 (GraphPad Software Inc., La Jolla, CA, USA). One-way ANOVA followed by Tukey’s post hoc test and column statistics were used for comparisons (* *p*  ≤  0.05; ** *p*  ≤  0.01; *** *p*  ≤  0.001; **** *p*  ≤  0.0001 were considered statistically significant).

## 3. Results

### 3.1. UT-SCC-24B and UT-SCC-42B Monocultures and Cocultures in µ-Plate Angiogenesis 96-Well Plate

Firstly, to determine the interaction and optimal cell-culture conditions for CAFs and HNSCC, a series of experiments with cell monocultures and cocultures was performed. The effect of the tumor to CAFs ratio on the morphology of three-dimensional organoids that form in 3D monocultures and cocultures of UT-SCC-24B and UT-SCC-42B tumor cells with hTERT-immortalized and GFP-labeled CAFs after 5 days of seeding is represented in Figure 2. The monocultures of UT-SCC-24B and UT-SCC-42B tumor cells resulted in the formation of well-defined, rounded, and similar-sized organoids when applying the sandwich model. The stepwise increase of CAF numbers or ratio resulted in increasingly irregular tumor organoids of reduced size, connected by strings of fibroblasts. Appendix A show UT-SCC-42B cells with GFP-labeled CAFs in 3D mono-and coculture in the sandwich model.

### 3.2. Replacing hTERT-Immortalized and GFP-Labeled CAFs and Moving from µ-Plate Angiogenesis 96-Well Plate to Costar^®^ 12-Well Cell Culture Multiple-Well Plate

To further refine our complex model system, changes were made to the 3D cell-culturing approach and the microtiter plate type used. Even though hTERT-immortalized and GFP-labeled CAFs showed benefits in facilitating visualization, they were derived from prostate tumor tissue, while this project aims to recapitulate the authentic tumor microenvironment of HNSCC. Therefore, prostate CAFs were replaced with patient-derived primary and unlabeled CAFs from HNSCC biopsies. The identity of unlabeled, primary patient-derived CAFs was tested by using IF staining with an anti-vimentin antibody. Furthermore, for molecular techniques such as qPCR and Western blotting, the small scale of the µ-Plate Angiogenesis 96-Well Plate was not sufficient and therefore replaced with 12-well Cell Culture Multiple-Well Plates. To allow side-by-side comparisons, both the 3D sheet models were performed in parallel in 12-well plates. While the 3D sandwich model resulted in the formation of increasingly irregular tumor organoids connected by strings of fibroblasts, the 3D sheet model resulted in the more rapid formation of large three-dimensional, tissue-like structures but did not feature organoids. Even though the 3:1 ratio resulted in the formation of robust 3D cocultures, the 1:1 ratio of cocultures with CAFs in both UT-SCC-24B and UT-SCC-42B in the 3D sandwich model resulted in the increased attachment of “runaway” tumor cells on the bottom of plastic wells. These cells were actively penetrating the lower gel in the sandwich model, adhered to plastic, and seriously disturbed 3D culture settings, imaging, and gel integrity, which was not the case in the 3D sheet model. The 3D sheet model delivered more robust, reproducible results, and was suitable for maintaining longer-term cell and tissue cultures, at significantly higher cell densities. Despite its increased density, imaging tissue-like structures and CAFs in the 3D sheet model by confocal microscopy was not compromised. The comparison of the 3D sandwich model and 3D sheet model in 3D mono- and cocultures using the UT-SCC-24B and UT-SCC-42B tumor cells with primary CAFs 5 days after seeding is shown in Figure 3 and Figure 4. Phase-contrast imaging (EVOS) was used for Figure 3 and 3D cube images generated by merging stacks of images generated by confocal microscopy are shown in Figure 4.

### 3.3. Immunofluorescence (IF) Staining

The 3D sheet model was our preferred option for testing drug chemosensitivity in HNSCC microtissues. Even though the 3D sandwich model has several advantages related mainly to facilitating real-time, life cell imaging and monitoring morphologic changes in organoids, it also has limitations. In particular, 3D cultures that contain a large ratio of highly active patient-derived CAFs lead to contraction of the gel after 5 days and often result in attachment of the cells on the bottom of plastic surfaces. IF staining of UT-SCC-24B and UT-SCC-42B cells in 3D mono- and cocultures with CFs were performed for the epithelial-specific antigen e-cadherin (carcinoma cells, organoids, and tissue-like aggregates) and the mesenchymal marker vimentin (VIM; CAFs and normal fibroblasts), counterstained with the DNA-dye Hoechst 33342 (Figure 4). The 3D cube images generated by merging a stack of individual stack images from confocal microscopy show the depth of the tissue-like 3D structures spontaneously forming after 5 days by the interaction of tumor cells with CAFs. The 3D cube images taken from the top of the model are represented in more detail in Appendix A, and Appendix A (UT-SCC-42B cells in 3D mono- and coculture in the 3D sheet model).

### 3.4. qRT-PCR Analysis for 3D Monocultures and Cocultures with Different Tumor/CAF Ratio

The expression of NOTCH1 and NOTCH3 mRNA and fold changes in monocultures of UT-SCC-42B, UT-SCC-24B, and CAFs, compared to 3D cocultures of different tumor cells with varying ratios to CAFs (3:1, 1:1, 1:3), was performed by qRT-PCR. It was observed that the UT-SCC-42B cell line shows the highest NOTCH1 mRNA expression, while patient-derived CAFs have higher NOTCH3 mRNA levels than UT-SCC-24B, similar to UT-SCC-42B (Figure 5). A high ratio of CAFs to tumor cells in 3D cocultures leads to an increase in NOTCH3 expression in cocultures of both UT-SCC-24B (Figure 5) and UT-SCC-42B cells (Figure 5). The presence of CAFs did not usually affect the NOTCH1 expression levels, except in the coculture with UT-SCC-24B cells in the 3:1 ratio, which only slightly increased (Figure 5). In summary, the presence of CAFs generally and most significantly increased the expression level of NOTCH3 mRNA but not NOTCH1 in all CAFs/HNSCC cell cocultures. This reproducible and marked effect is positively correlated with the number of CAFs used in cocultures.

### 3.5. WST-8 Assays

To test whether inhibition of Notch signaling by specific inhibitors (such as CRE) will affect the sensitivity of HNSCC cells (with and without CAFs) to cytostatic drugs including cisplatin (CDDP), after 5 days of culture, the 3D culture of UT-SCC-24B and UT-SCC-42B cell lines with patient-derived CAFs (ratio 1:1) were treated with cisplatin alone or in combination with crenigacestat. After 2 days of drug exposure, the WST-8 metabolic assay was performed. Monocultures of UT-SCC-42B were significantly more sensitive to cisplatin than monocultures. Furthermore, UT-SCC-24B cells were more resistant than UT-SCC-24B cells (Figure 6A,B). The addition of CAFs in 3D cocultures resulted in similar sensitivity scores compared to treatments in 3D monoculture. Upon combining CRE with cisplatin treatment, the two cell lines behaved in opposite directions (Figure 6). Thus, blocking Notch signaling promoted the differences in the chemosensitivity of both cell lines.

### 3.6. Western Blot Analysis

To confirm and verify the qPCR data regarding the increase in activation of Notch signaling by CAFs in HNSCC/CAFs cocultures, we also analyzed protein expression. Western blotting was performed for NOTCH1, NOTCH3, and JAG2 proteins after treating mono- and cocultures of UT-SCC-24B and UT-SCC-42B cells with patient-derived CAFs, using cisplatin alone, or combined 5 μM cisplatin with 5 μM crenigacestat. In addition, expression of the NOTCH3 protein was compared between mono- and cocultures of UT-SCC-24B and UT-SCC-42B cells, and patient-derived CAFs alone. Results from Western blotting are shown in Figure 7 and, in addition, quantification of differences in protein expression using densitometry is shown in Appendix A. For both UT-SCC-24B and UT-SCC-42B cells, an increase in the expression of N1ICD, the full length of NOTCH3 protein, and N3ICD was observed in CAFs/HNSCC cocultures compared to tumor monocultures, confirming the qPCR results. Interestingly, in 3D monocultures of both cell lines, exposure to CDDP increased the expression of both N1ICD and N3ICD. However, co-treatment of CDDP and CRE cocultures of both UT-SCC-42B and UT-SCC-24B cells with CAFs resulted in opposing effects on NOTCH1 compared to NOTCH3 proteins: with the co-treatment, we consistently observed an efficient decrease in the expression of NOTCH3 and N3ICD, but no changes in NOTCH1 and N1ICD expression. Finally, for UT-SCC-24B cells, an increase in the expression of JAG2 protein was observed in cocultures of CAFs and HNSCC cells, compared to monocultures. In contrast, this effect was not observed for the UT-SCC-42B cell line. Additionally, JAG2 levels decreased after treatment with both CDDP and CDDP/CRE combinations in both tested cell lines. The densitometry analyses for all Western blots assembled in Figure 7 are shown in Appendix A. Full versions of Western blots have been submitted for review. Taken together, our data show that the presence of CAFs differentially activates Notch signaling in 3D cocultures, which modifies the drug sensitivity to cisplatin. This demonstrates the potential of combining CDDP and CRE treatment to reduce Notch signaling and may be useful to increase the chemosensitivity of head and neck cancer cells to cisplatin in some patients, depending on Notch signaling activity.

## 4. Discussion

Traditional two-dimensional (2D) monolayer cultures on glass or polystyrene surfaces have been widely used to test anti-cancer drugs for decades. However, these simple but utterly reductionist cell cultures have a wide range of limitations. They fail to mimic the basic physiological conditions found in tumor tissues, including, among others: (a) the lack of proper cell–cell and cell–matrix interactions, (b) the absence of any mechanical properties prevalent in tumor tissues, (c) inadequate representation of metabolic gradients, (d) the lack of oxygen gradients and hypoxia [41,42,43], (e) acquiring a forced apical–basal polarity not found in tumor cells embedded within cancer tissues, and (f) as a consequence, they fail to mimic the characteristic tumor histology or architecture. As a result, 2D monolayer and monoculture models often exhibit marked hypersensitivity to cytostatic drugs which are preferentially targeting hyperproliferation and induce growth arrest and/or programmed cell death/apoptosis. This bias is leading to inaccurate drug sensitivity scores and compromised assessment of drug resistance, e.g., in early-stage drug discovery and preclinical studies. We also hypothesized that routine 3D organoid models in a laminin-rich matrix that lack stromal counterparts (=fibroblasts, CAFs) are likely not fully representative of addressing these questions. To experimentally test the benefit of more complex, more organotypic, or more tissue-like model systems, we have chosen 3D cultures of two selected HNSCC tumor cells (cell lines UT-SCC-24B and -42B) that differ in sensitivity to cisplatin. These are first cocultured with hTERT-immortalized and GFP-labeled CAFs either in 96-well angiogenesis plates (optimized for confocal imaging) or, later, with patient-derived primary CAFs in 12-well plates. To allow a direct side-by-side comparison between these models, we have used two different 3D coculture settings: the previously described “sandwich model” [26,27,28] and the “3D sheet model” which is described by us in this manuscript. For consistency, both models utilize the same 3D matrix that consists of a 2:1 mix of Matrigel and collagen type I (2 mg/mL of Matrigel and 1 mg/mL of collagen), which represents an equal 1:1 mix of their protein concentration.

In the initial investigation, we explored the influence of varying numbers and ratios of UT-SCC-24B and UT-SCC-42B tumor cells to CAFs on the formation of organoid structures in µ-angiogenesis plates. We also utilized pure Matrigel, a laminin-rich basement membrane extract derived from mouse xenografts, which strongly promotes the growth and differentiation of epithelial cells, including squamous carcinoma cells. Specifically, laminin-rich matrices like Matrigel promote the formation of well-differentiated and polarized organoids, even from transformed cancer cells and cell lines [26]. However, Matrigel does not support the growth of stromal components, such as normal fibroblasts and CAFs. In contrast, collagen type I, a major constituent of the normal connective tissue and highly enriched in the tumor matrix, provides a favorable environment for the growth and functional differentiation of mesenchymal or stromal cells, such as normal fibroblasts and CAFs. Simultaneously, it promotes the growth of most tumor cell lines—but typically not in the form of organoids since differentiation is not promoted. As a consequence, pure Matrigel matrix can lead to the rapid loss of stromal cells, while pure collagen type I gels can result in the excessive proliferation of CAFs and gel contraction within a few days. To address these challenges, we evaluated a 1:1 mixture of Matrigel and collagen type I, which simultaneously supports both tumor cells and CAFs, does not result in contraction, and still promotes organoid formation. Embedding single tumor cells into Matrigel/collagen gels using the sandwich model results in the formation of well-defined, rounded, and evenly sized organoids. These organoids are similar to those grown in pure Matrigel and largely clonal, i.e., each organoid originates from a single tumor cell. Adding a relatively small number of cancer-associated fibroblasts (ratio tumor/stroma 3:1) results in the forming of more irregularly shaped tumor organoids or aggregates. Increasing numbers of fibroblasts (ratio 1:1 or 1:3) result in even more irregular and unevenly sized tumor organoids, connected by strings of fibroblasts. At a later time point of 3D coculture (day 5+) in the sandwich model, larger numbers of CAFs also result in the contraction of the gel.

Three-dimensional cultures of both UT-SCC-24B and UT-SCC-42B form very similar multicellular structures (organoids) but differ significantly in drug sensitivity. To be able to extract a sufficient amount of protein and mRNA for subsequent analyses, we transferred our experiments to ordinary 12-well microtiter plates instead of the specialized, imaging-ready µ-angiogenesis 96−well plates. Simultaneously, we replaced the hTERT-immortalized and GFP-labeled CAFs (originally derived from [30]) with patient-derived primary CAFs from HNSCC, to better mimic the microenvironment of squamous epithelial cancers. The 3D sandwich model facilitates real-time confocal imaging of 3D cultures, making it suitable for rapid, miniaturized high-content, or imaging-based analyses, especially in early-stage drug discovery and compound screening [44], and distinction of tumor versus stromal cells. However, it has limitations for subsequent functional analyses and molecular biology studies: (a) highly active patient-derived CAFs eventually contract the gel and tend to attach to the plastic bottom even after 5 days, especially with high CAF/tumor cell ratios (1:1 and 1:3); (b) the long-term viability of ultra-thin, miniaturized microtissues is low [44]; and (c) it does not provide sufficient cell numbers for subsequent RNA or protein extraction required in follow-up molecular analyses. Simply upscaling the assay format and the size of gels, however, posed a new challenge as the larger gels showed even more compromised long-term integrity and stability.

In this context, we made an intriguing observation regarding the spontaneous generation of flat 3D sheets of living tissue-like structures, which we refer to here as the “3D sheet model”. The fashion of how tumor and stromal cells were seeded on top of the gel had a notable impact on the rapid and efficient formation of large-scale, tissue-like structures in this model. It also did not result in the distinct formation of organoids, which are not necessarily representative of the histology of squamous carcinomas. When a smaller number of cells are seeded, they also spontaneously organize themselves in tumor “islands” surrounded by stromal areas (work in progress, to be published in a follow-up manuscript). Specifically, we observed a substantial increase in cell proliferation, cell motility, and the rapid formation of tissue-like tumor masses that contained both tumor and stromal cells. These complex 3D structures also showed an active expansion and penetration of cells into the gel matrix, resulting in a significantly thicker layer of relatively dense tissue that provided additional stability, e.g., for long-term drug exposures. Notably, we did not observe any gel contraction mediated by fibroblasts, regardless of their numbers; nor did we observe the characteristic escape of tumor cells through the gel to the plastic bottom, as frequently observed in the sandwich model.

Both tumor cell lines, UT-SCC-24B and -42B, exhibited a morphology in the 3D sheet model that closely resembles the characteristic cribriform histopathology often seen in aggressive HNSCC and adenocarcinomas. When using a 1:1 ratio of tumor cells to CAFs, there was a uniform distribution of cells, allowing for enhanced tumor–stroma interaction. The sandwich method gave flexibility to cells to move inside of the gel or to sit on top of the gel. In contrast, the 3D sandwich model predominantly produced polarized organoids with varying levels of squamous differentiation, depending on the cell lines utilized. These show restricted cell movements compared to the 3D sheet model unless extremely aggressive tumor lines are used that may skip the formation of organoids. The same applies to non-epithelial tumor types such as melanoma or glioblastoma. In contrast, when the same tumor cells and CAFs were embedded uniformly in the Matrigel/collagen gel used for the 3D sheet models they tended to form small, slowly growing organoids surrounded by CAFs similar to the sandwich model and also yielded limited numbers of cells for subsequent molecular analyses. The cribriform structure observed upon “on top” seeding of tumor cells in the 3D sheet model is characterized by the formation of prominent openings or cavitations that stretch between epithelial areas of proliferation, with monomorphic round cell populations often observed between and inside these holes. These cribriform-like structures may form as the result of tension that rapidly forms between tumor cells and CAFs, thus indicating intense and dynamic cell–cell interactions. The cribriform-like morphology may be the sign of aggressive HNSCC cells and has been previously associated with hypoxic growth conditions, altered epigenetic regulation, and poor prognostic outcomes in certain cancers [45,46,47]. The 3D sheet model also demonstrates increased cell density, not only reducing the risk of early gel dissociation but also exhibiting intrinsic resistance against anti-cancer drugs. This is likely due to dense tissue-like layers of cells and cell membranes, which can act as barriers to drug delivery [48] and penetration—probably more similar to the conditions in cancer tissues than organoid models. These characteristics led us to select the “3D sheet model” for all subsequent qRT-PCR and drug testing with subsequent molecular analysis, focusing on the relevance of the Notch signaling pathway in HNSCC.

In 3D monoculture, UT-SCC-42B cells exhibited the highest NOTCH1 mRNA levels, while patient-derived CAFs demonstrated the highest NOTCH3 levels, closely followed by UT-SCC-42B. Both 3D mono- and cocultures of UT-SCC-24B showed consistently low NOTCH1 and NOTCH3 expressions. In 3D cocultures of both UT-SCC-42B and UT-SCC-24B, there was a noticeable increase in NOTCH3 expression as the number of CAFs increased. This is possibly related to recent observations showing that metastatic (and drug-resistant) HNSCC lesions may develop a dependency on NOTCH3 (over-)expression, partly overcoming the initial loss of NOTCH1 observed in many tissues [49]. Our observation, however, relates to the CAFs contributing to this change, not the tumor cells themselves.

In contrast, expression of NOTCH1 protein was not significantly induced by coculture with CAFs. This led us to conclude that NOTCH3 may play a more critical role in the formation of tissue-like microtumor structures and likely cisplatin resistance, and that this increased NOTCH3 protein expression is mainly attributed to the contribution of CAFs. We also speculate that our modified approach may have resulted in the formation of a partial air–liquid interface (ALI), generating an effective oxygen gradient within the forming sheet-like microtissues. Oxygen gradients and an ALI have been previously reported to support tissue formation, cell differentiation, and homeostasis, full validation of which in our model system will require additional experiments (ongoing research). The advantage of the 3D sheet model may thus preferentially lie in the localized formation of hypoxia or oxygen gradients, which typically occur in structures located at a distance of more than 300 μm from the primary oxygen source, such as capillaries. In addition, the formation of high cell density and tissue-like structures that impede drug delivery and penetration likely contributes to increased resistance against cytotoxic compounds [50] like the widely used cisplatin.

In HNSCC, only a few chemotherapeutic drugs are approved for chemotherapy or combined chemoradiotherapy. These include platinum-based drugs like cisplatin and carboplatin, the nucleotide analog 5-fluorouracil (5-FU), taxanes such as docetaxel and paclitaxel, and, occasionally, methotrexate [5,6]. However, drug resistance can develop against all these compounds in usually advanced patients undergoing chemotherapy. Up to 50% of patients treated with cisplatin experience acquired drug resistance [10]. For this reason alone, it is worthwhile to explore the potential synergistic effects of common standard-of-care drugs with novel, targeted agents that impact “classic” cancer-related signaling like NOTCH signaling. This approach may potentially uncover new combinatorial effects on drug chemosensitivity. Even more importantly, this approach may eventually become an important aspect of personalized or individualized medicine, where specific drug vulnerabilities and even synthetic-lethal effects are tested on a patient-by-patient basis. NOTCH1 has been identified as one of the most frequently mutated genes in HNSCC, with inactivating or loss-of-function (LoF) mutations present in 10% to 15% of cases reviewed in [51]. Even patients without genetic mutations in NOTCH receptors may exhibit very low expression levels of NOTCH1 due to epigenetic inactivation. Additionally, differential expression levels of other receptors like NOTCH2, NOTCH3, and Notch ligands like JAG1 and 2 are also observed in HNSCC [49]. However, the role of Notch signaling in HNSCC remains controversial and may not include only LoF mutations. In contrast, there is increasing evidence that, especially in advanced cases (relapsed and metastatic HNSCC), the tumor suppressor function of NOTCH receptors—which is highly predominant and defining in early-stage cancers—may be reversed for the opposite, exerting oncogenic activation [13]. This may further include functional compensation between the four NOTCH receptors, especially in advanced, aggressive tumor cells, although this potential is currently very poorly understood and under-researched. Only recently, evidence for a potential compensation between NOTCH1 and NOTCH3, with an increased dependence of metastatic HNSCC on NOTCH3 overexpression, has been reported [15]. Functional compensation may overcome the loss of NOTCH1 in the early stages of cancer initiation and recovery of Notch signaling activity, thus acquiring a potentially oncogenic role that may promote tumor progression, relapse, invasion, tumor cell plasticity, and drug resistance—similar to the situation observed in TNBC. This may also be promoted by the simultaneous overexpression of Notch ligands like JAG1 and JAG2, which are observed in advanced HNSCC. Notch signaling further acts as a mechanoreceptor, with the intensity of signaling being proportional to the area between cells in direct contact [52]. Therefore, the shift from 2D to 3D models and increasingly dense microtissues is likely to result in a significant activation of Notch signaling. We postulate that implementing strong contacts to stromal CAFs will further modulate the role of Notch signaling in such cancer models. Also, the formation of complex tissue-like structures in 3D models, which goes beyond the formation of simple, but partially de-differentiated organoids, will likely affect Notch signaling and the response to Notch inhibitors and/or cisplatin. In a recent study, 45 HPV-negative head and neck carcinoma cell lines were investigated for drug sensitivity against 250+ drugs, including cisplatin [31]. Our tests confirmed that UT-SCC-24B indeed shows higher resistance toward cisplatin than UT-SCC-42B, according to the 2D studies by [31]. Next, we investigated the combinatorial effects of cisplatin and gamma-secretase/Notch-inhibitor crenigacestat in 3D monocultures of both UT-SCC-24B and UT-SCC-42B cells, in comparison to 3D cocultures of these cells with CAFs. Adding CAFs in 3D coculture to the more sensitive UT-SCC-42B cell line reproducibly lowered the drug sensitivity of this line. In the less sensitive UT-SCC-24B cells, the opposite was observed. In both cases, the presence of increasing numbers of CAFs significantly altered the drug sensitivity in comparison to 3D monoculture. We hypothesize that these effects may be due to altered Notch signaling activity in the complex microtissues and mediated by the CAFs.

It was suggested that high levels of NOTCH1 and NOTCH3 protein expression, even without any treatment, may correlate with cisplatin resistance. Crenigacestat is an oral Notch and gamma-secretase inhibitor that was reported to specifically block the expression and activity of NOTCH1 [14,18,53]. We were particularly interested in determining whether Notch signaling, as a stress-response-related pathway, may be specifically affected by cisplatin treatment and play a role in acquired drug resistance. We initially hypothesized that high levels of NOTCH1 protein expression before treatment (both the full-length protein and the N1ICD) might generally indicate increased drug sensitivity [14,54]. However, our findings showed that the UT-SCC-24B cell line, which exhibited significantly lower NOTCH1 mRNA and protein expression than UT-SCC-42B, was still more resistant to cisplatin compared to UT-SCC-42B, regardless of the NOTCH1 levels. In the 3D monoculture of UT-SCC-24B and coculture with CAFs, both NOTCH1 and NOTCH3 protein expression notably increased with cisplatin treatment. The more cisplatin-sensitive UT-SCC-42B monocultures displayed a similar, albeit milder increase. However, cocultures of these cells with CAFs exhibited an unexpectedly sharp decline in both NOTCH1 and NOTCH3 levels. We concluded that there may be a correlation between cisplatin resistance and increased or inducible levels of NOTCH1 and NOTCH3 protein expression, regardless of their initial levels before cisplatin treatment, but it depends on the presence or absence of CAFs.

We assumed that the presence or absence of fibroblasts in tumor/stroma cocultures may play a decisive role in cisplatin resistance, partly because of their role in Notch signaling. In our experiments, crenigacestat was not exclusively linked to NOTCH1 functions but, instead, primarily decreased NOTCH3 expression levels and only slightly decreased NOTCH1 activation. This observation may relate to the specific inhibition of NOTCH3 expression and functionality in CAFs by the drug, especially in 3D cocultures and after treatment with cisplatin. Finally, we analyzed the expression of the Notch ligand JAG2 expression in 3D cultures. We observed a clear decrease in the JAG2 level after cisplatin exposure, which correlated with the WST-8 viability test results for both 3D monocultures and cocultures of both tumor cell lines. We initially expected that JAG2 expression might correlate with decreased cell viability and increased tumor cell death, but further research is needed to confirm this.

## 5. Conclusions

The advancement in understanding the microenvironment of cancer tumors requires a shift from traditional two-dimensional (2D) models to more complex three-dimensional (3D) culture systems. Our study underlines the shortcomings of 2D monolayer cultures, especially in drug sensitivity testing, and emphasizes the need for physiologically relevant model systems. Using HNSCC cell lines with differential chemosensitivity to cisplatin, we experimented with drug testing with various 3D culture conditions and identified the “3D sheet model” as a superior and informative methodology. This model more closely mimics the TM and tumor histology, by capturing the complex interactions between tumor cells and their stromal counterparts. We are convinced this approach facilitates a more accurate drug response assessment.

Notably, the 3D sheet model displayed characteristic tumor morphologies, such as the cribriform structures, suggesting its potential to recreate aggressive HNSCC conditions. These structures, in combination with high cell density and extracellular matrix components, can significantly affect drug penetration and efficacy, thus impacting chemosensitivity. Our findings on the Notch signaling pathway, especially the inducible, increased NOTCH3 expression in cocultures, indicate its critical role in tumor development and possibly acquired drug resistance, according to recent findings. The shift from 2D to 3D cultures undoubtedly influences Notch signaling and understanding this modulation can pave the way for targeted therapeutic strategies.

While cisplatin remains the first preferred therapy in HNSCC, resistance remains a problematic issue. Our study digs into the potential synergies between cisplatin and Notch inhibitors, emphasizing the relevance of combinatorial approaches. By investigating the interplay between the tumor microenvironment, Notch signaling, and drug response in 3D models, we are coming closer to a more personalized and effective approach to cancer therapy. Future research should aim to validate these findings in a larger perspective and explore other potential drug combinations to expand our therapeutic arsenal against aggressive HNSCC.

## Figures and Tables

**Figure 1 cancers-15-05320-f001:**
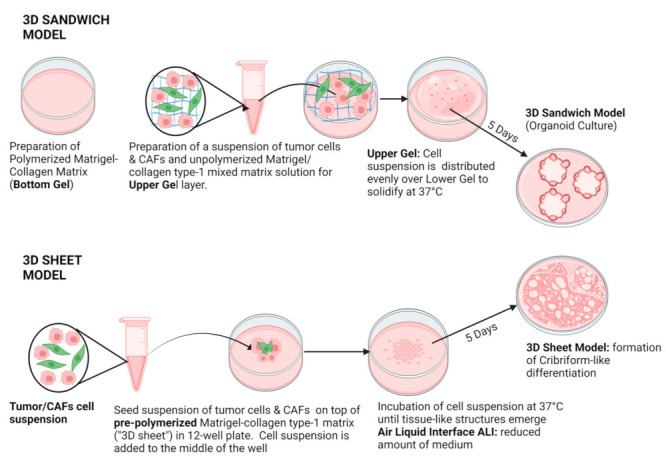
Schematic representation of different 3D culture and coculture methods. Top row: the sandwich model, bottom: 3D sheet model. The sandwich model uses a simplified, well-in-a-well design to minimize the meniscus formation in liquid media, which severely affects automated confocal microscopy but at the same time represents a highly miniaturized model system in which a small number of organoids emerge between two layers of matrix. In contrast, the 3D sheet model is generated in 12-well plates instead of 96-well plates and can generate a significantly larger amount of biomaterials for subsequent molecular analyses. It further changes the geometry of the model system, and allows a larger interface with media and air, resulting in strikingly different morphologies that spontaneously emerge upon coculture of tumor cells and fibroblasts.

**Figure 2 cancers-15-05320-f002:**
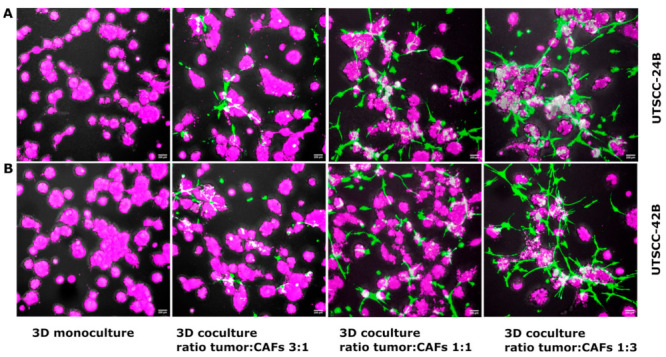
Coculture of tumor and stroma cells (CAFs) in the “sandwich model”. (**A**) Organoids formed by monoculture (left) and 3D coculture of UT-SCC-24B tumor cells with hTERT-immortalized and GFP-labeled CAFs (green), seeded between 2 layers of Matrigel/collagen-type-I matrix (2 mg/mL of Matrigel and 1 mg/mL collagen) at different ratios into each well of an IBIDI µ-angiogenesis plate. The combined total number of cells seeded per well does not exceed 3000 per well, with 2250 tumor cells/750 CAFs for a ratio of 3:1, 1500 tumor cells/1500 CAFs for a ratio of 1:1, and 750 tumor cells/2250 CAFs for a ratio of 1:3. (**B**) Monoculture (left) and 3D coculture of UT-SCC-42B tumor cells with hTERT-immortalized and GFP-labeled CAFs, seeded into the same Matrigel/collagen-type-I matrix (2 mg/mL of Matrigel and 1 mg/mL collagen), and identical cell numbers/ratios to those described for UT-SCC-24B. Tumor cells and the emerging 3D organoids were labeled with Vybrant™ DiD cell-labeling solution (pink) directly before seeding into the wells. Confocal images of organoids were taken on day 5 after seeding into the gels, at 200× magnification with a NIKON confocal microscope. Scale bars indicate 200 µm.

**Figure 3 cancers-15-05320-f003:**
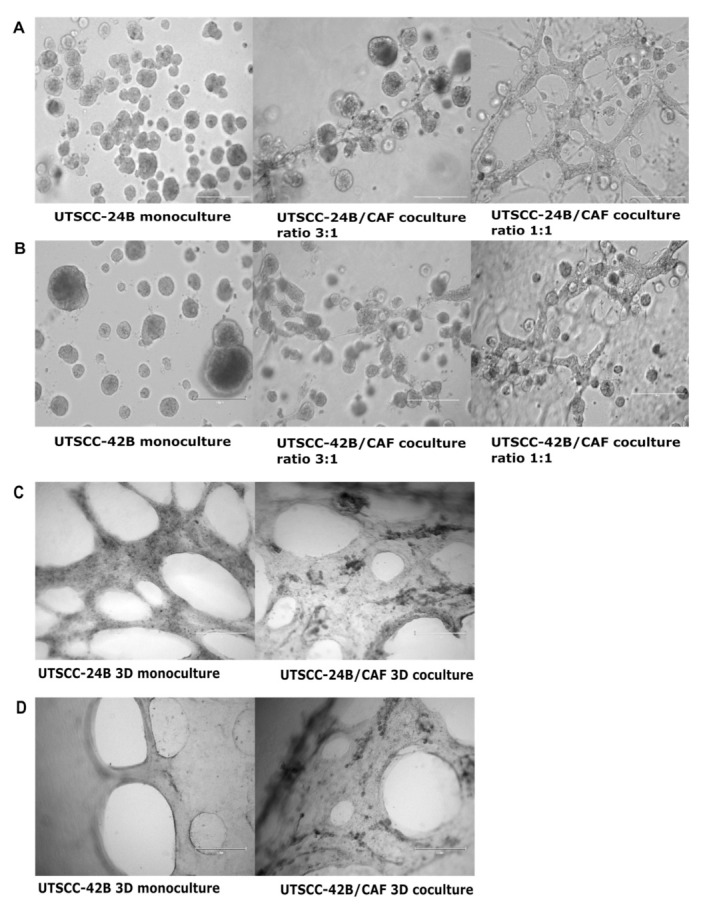
Comparison of 3D mono- and cocultures in the 3D sandwich models with the 3D sheet model in Matrigel/collagen type-I mixed matrix. (**A**) Left: Phase-contrast images of 3D monoculture of UT-SCC-24B cells embedded in the 3D sandwich model, Middle: Phase-contrast microscopy images of 3D coculture of UT-SCC-24B cell line with patient-derived primary CAFs at the 3:1 ratio. Right: Same with CAFs at a 1:1 ratio. (**B**) Left: 3D monoculture of UT-SCC-42B cells in the 3D sandwich model. Middle: UT-SCC-42B cell line with patient-derived CAFs at a 3:1 ratio. Right: same cells with CAFs at a 1:1 ratio. EVOS phase-contrast images were taken on day 5 after seeding cells at 100× magnification. Scale bars indicate 250 µm. (**C**) Three-dimensional monoculture and coculture of UT-SCC-24B cancer cells and CAFs (1:1 ratio) on top of a Matrigel/collagen-type-I gel, the “3D sheet model”. Left: 3D organoid monoculture of UT-SCC-24B cells in the 3D sheet model. Right: 3D coculture of UT-SCC-24B cell line and CAFs (1:1 ratio). (**D**) Left: 3D monoculture of UT-SCC-42B cells and Right: 3D coculture of UT-SCC-42B cell line and CAFs. EVOS images were taken on day 5 after seeding cells at 40× magnification. Scale bars indicate 600 µm.

**Figure 4 cancers-15-05320-f004:**
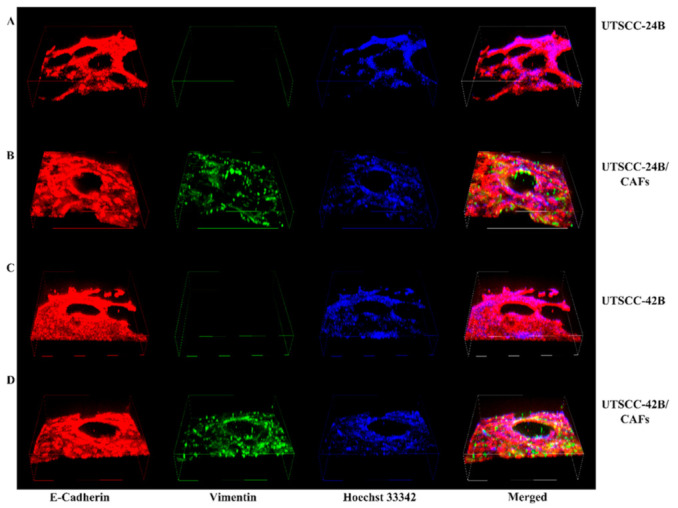
Analysis of multicellular, tissue-like aggregates forming in 3D coculture of tumor cells (UT-SCC-24B and 42B) with patient-derived CAFs in the “3D sheet model”. (**A**) Three-dimensional monoculture of UT-SCC-24B tumor cells, Width (W): 1290.67 μm, Height (H): 1290.67 μm, Depth (D): 378.10 μm. (**B**) Cocultures of UT-SCC-24B tumor cells and patient-derived fibroblasts (1:1 ratio), W: 1290.67 μm, H: 1290.67 μm, D: 267 μm. (**C**) Monocultures of UT-SCC-42B tumor cells, W: 1290.67 μm, H: 1290.67 μm, D: 333.28 μm. (**D**) Cocultures of UT-SCC-42B tumor cells and patient-derived fibroblasts (1:1 ratio), W: 1290.67 μm, H: 1290.67 μm, D: 408.63 μm. All structures formed on the surface of a Matrigel/collagen type I mixed gel. The red color represents E-cadherin, green vimentin, and blue Hoechst 33342. Confocal images were captured on day 5 post-seeding using a NIKON confocal microscope at 100× magnification in the cube version.

**Figure 5 cancers-15-05320-f005:**
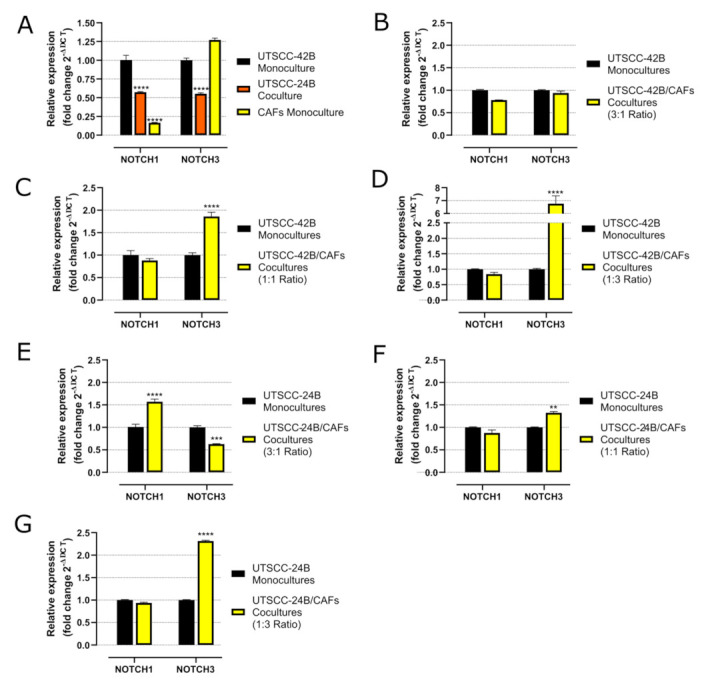
mRNA fold change measured by qRT-PCR for NOTCH1 and NOTCH3 with different ratios of patient-derived CAFs. (**A**) Comparison of NOTCH1 and NOTCH3 expression in 3D monocultures of UT-SCC-24B, UT-SCC-42B, and CAFs in the 3D sheet model. (**B**) Comparison of NOTCH1 and NOTCH3 expression in monocultures of UT-SCC-42B versus cocultures with CAFs at the 3:1 cell ratio. (**C**) Same comparison for the 1:1 tumor to CAFs ratio and (**D**) for the 1:3 tumor to CAFs ratio. (**E**) Comparison of NOTCH1 and NOTCH3 signaling in 3D monocultures of UT-SCC-24B cells versus 3D cocultures with CAFs for the 3:1 tumor to CAFs ratio. (**F**) Same comparison for the 1:1 tumor to CAFs ratio and (**G**) for the 1:3 tumor to CAFs ratio, all in the 3D sheet model. Error bar represents ± SEM for *n =* 3. Statistically significant differences are indicated by stars ** *p*  ≤  0.01; *** *p*  ≤  0.001; **** *p*  ≤  0.0001.

**Figure 6 cancers-15-05320-f006:**
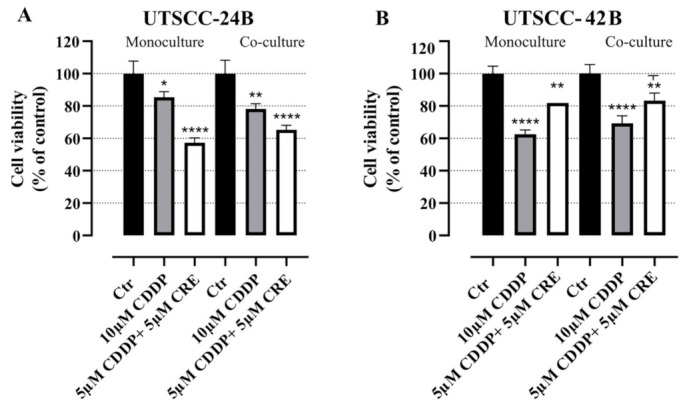
Sensitivity of 3D cultures and cocultures against cisplatin and cisplatin combination with the Notch-pathway inhibitor crenigacestat. (**A**): WST-8 assay results of monocultures and cocultures of UT-SCC-24B tumor cells with patient-derived fibroblasts without any treatment followed by cisplatin treatment and combination treatment with cisplatin (CDDP) and crenigacestat (CRE). (**B**): WST-8 assay results of the same treatments of 3D monocultures and cocultures of UT-SCC-42B cells. Ctr (M) 3D Monoculture control; Ctr (C) 3D Coculture control. The black color represents control groups without any drug treatment, the grey color represents only CDDP treatment, white color represents the combinatory treatment of CDDP and CRE. * *p*  ≤  0.05; ** *p*  ≤  0.01; **** *p*  ≤  0.0001 vs. Ctr (M)/(C) was considered statistically significant. Error bar represents ± SD for *n* = 3.

**Figure 7 cancers-15-05320-f007:**
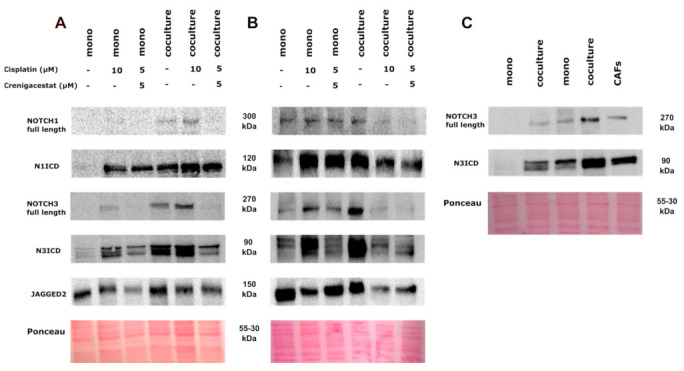
Western-blot analysis of Notch-pathway-related protein expression. (**A**) From left to right: monoculture of UT–SCC–24B without any treatment, versus treated with 10 μM cisplatin (CDDP), and 5 μM CDDP + 5 μM crenigacestat (CRE), 3D coculture of UT–SCC–24B with patient-derived fibroblasts, without any treatment, treated with 10 μM CDDP, and treated with 5 μM CDDP + 5 μM CRE. Blots were tested with antibodies against NOTCH1/FL–300 kDa, NOTCH1/N1ICD–120 kDa, NOTCH3/FL–270 kDa, NOTCH3/NICD–90 kDa, and JAG2–150 kDa. (**B**) From left to right: 3D monoculture of UT–SCC–42B without any treatment, treated with 10 μM CDDP, 5 μM CDDP + 5 μM CRE, coculture of UT–SCC–42B with CAFs, treated with 10 μM CDDP, treated with 5 μM CDDP + 5 μM CRE. Blots were tested for NOTCH1/FL–300 kDa, NOTCH1/NICD–120 kDa, NOTCH3/FL–270 kDa, NOTCH3/NICD–90 kDa, and JAG2–150 kDa. (**C**) Controls of UT–SCC–24B, UT–SCC–42B, and patient-derived fibroblasts (CAFs) for NOTCH3/FL–270 kDa and NOTCH3/NICD–90 kDa.

## Data Availability

The data presented in this study are available in the presented article and Appendix A. Any additional data related to this study are available on request from the corresponding author.

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
