# Peer review of "Optimization of a Three-Dimensional Culturing Method for Assessing the Impact of Cisplatin on Notch Signaling in Head and Neck Squamous Cell Carcinoma (HNSCC)"

_cancers, 2023, doi:10.3390/cancers15225320_

Round 1

Reviewer 1 Report

Comments and Suggestions for Authors

This is an interesting paper where the purpose of the authors was to optimize a 3D cell culture method. They tested two HNSCC cell lines with different cisplatin sensitivities. Moreover, the authors demonstrate that elevated NOTCH1 and NOTCH3 levels were consistently related to reduced cis-platin sensitivity and cell survival in 3D models. They conclude that 3D culture cells recreate aggressive HNSCC conditions.The study design is adequate to address the scientific question. The figures are adequate since the authors should improve the quality of the figures. Moreover, the authors should include the densitometric quantification of all the western blots.

Author Response

Reviewer 1

Comments and Suggestions for Authors

This is an interesting paper where the purpose of the authors was to optimize a 3D cell culture method. They tested two HNSCC cell lines with different cisplatin sensitivities. Moreover, the authors demonstrate that elevated NOTCH1 and NOTCH3 levels were consistently related to reduced cis-platin sensitivity and cell survival in 3D models. They conclude that 3D culture of cells recreate aggressive HNSCC conditions.

We are thankful for the appreciation by the reviewer. As the manuscript has a predominantly technical orientation, this is our main message, but we also have functional insights.

The study design is adequate to address the scientific question. The figures are adequate since the authors should improve the quality of the figures. Moreover, the authors should include the densitometric quantification of all the western blots.

Thanks also for these suggestions. In fact, we have noticed that the figures embedded in the pdf file (which was submitted for review) were indeed of low quality. We did not notice this by ourselves; probably since the MS Word document was submitted simultaneously in which there were no such quality issues. We have now fixed the quality/resolution problems, which most likely were related to converting .doc files into .pdf files. We apologize for this technical issues but we did not notice ourselves at the time of submission.

Densitometric quantification of all Western blots have already been added to the “Supplemental Data” file upon the 1st round of submission: This is currently Supplemental Figure 2 in the original version. We learn from the reviewer that it may be required to be more transparently referred to in the main text so it is not getting missed by reviewers and/or readers.

Reviewer 2 Report

Comments and Suggestions for Authors

This is an interesting study about optimization of 3D culturing method for assessing cisplatin's impact on Notch signalling in head and neck squamous cell carcinoma (HNSCC).

Two cell lines were used. The research aimed to optimize a 3D cell culture method for assessing the interplay between tumor cells and cancer-associated fibroblasts (CAFs) in vitro, and to study how cisplatin impacts the Notch pathway, particularly considering the role of CAFs.

The paper is well written. Materials and methods are adequately described.

Results and discussion are appropriate. References are adequate.

Author Response

Reviewer 2

Comments and Suggestions for Authors

This is an interesting study about optimization of 3D culturing method for assessing cisplatin's impact on Notch signalling in head and neck squamous cell carcinoma (HNSCC).

Two cell lines were used. The research aimed to optimize a 3D cell culture method for assessing the interplay between tumor cells and cancer-associated fibroblasts (CAFs) in vitro, and to study how cisplatin impacts the Notch pathway, particularly considering the role of CAFs.

The paper is well written. Materials and methods are adequately described.

Results and discussion are appropriate. References are adequate.

There is nothing to respond or amend according to the reviewer's comments here. We are grateful for the overall positive appraisal of our manuscript.

Reviewer 3 Report

Comments and Suggestions for Authors

please see attached review

Comments on the Quality of English Language

Attention to detail is required as detailed in the response document

Author Response

we responded to reviewer 3 in the pdf file uploaded below

Round 2

Reviewer 3 Report

Comments and Suggestions for Authors

The authors appear to have resolved the majority of the issues raised

Comments on the Quality of English Language

A careful check through once all corrections are removed is required

Author Response

Dear reviewer,

we would like to thank the reviewer once again for constructive comments, and contributing to a fair and productive reviewing process. We have now hopefully fixed all remaining English language issues, and accepted all changes done in round 1 into the consolidated manuscript.  Below a short summary of all the major issues that have been addressed, mainly in round 1, but also finalized now in round 2. Here is a short list of the most significant changes in 1st and 2nd remission:

  • We have added a graphical abstract and a short description for this abstract directly to the document.
  • We have slightly modified the abstract to highlight the molecular and functional findings, in addition to technical issues.
  • We have slightly changed the title: “Optimization of Organotypic 3D Cell Culture Methods to Assess the Impact of Cisplatin on Notch Signaling in Head and Neck Squamous Cell Carcinoma (HNSCC)” is now “Optimization of a 3D Cell Culture Method for Assessing the Impact of Cisplatin on Notch Signaling in Head and Neck Squamous Carcinoma”. We hope this change is acceptable.
  • We have tried to resolve the resolution and quality issues of all figures embedded in the text. This issue has emerged from the failure to convert MS Word files to PDF format without significant loss of resolution, especially of the embedded images (PNG format). We are happy to submit each ones of these figures, also including the supplemental figures, separately as SVG files or PNG (300 dpi resolution or more). We have also increased the resolution of Fig. 4 specifically.
  • We have expanded the introduction by adding a short description and rationale of the “3D sandwich model”, which was the starting point for developing our novel method. This also includes additional manuscripts in which this method and its previous applications in drug screening are described in detail.
  • Also, in the “introduction”, we further elaborated on the clinical standard-of-care therapies for head & neck cancer, as requested by reviewer 3; with a special highlight on cisplatin.
  • In both the “introduction” and the “discussion” sections, we have added recent references highlighting the emerging putative role of NOTCH signaling in the progression and acquired drug resistance of head & neck cancers against chemotherapeutic agents, especially, cisplatin.
  • The origin and the culture of the cell lines we have used in this manuscript has been described in more detail.
  • The methods section has been extended by a more detailed description of cell culture protocols, especially the culture and isolation of primary, cancer-associated fibroblasts from patient biopsies. We have also added references to elaborate on these methods. The same applies to the description of our IF staining protocol, which has been clarified now.
  • The three-dimensional nature of our tissue cultures, shown in Fig. 2, 3, and 4, has been further elaborated and clarified by adding four short videos to the manuscript's collection of supplemental data. These clearly show the 3D aspect of the cultures.
  • Figure 5, showing results from quantitative RT-PCR, has been completely remade.
  • Finally, the description and discussion of the Western blot data in the text has been refined, simplified, and hopefully, clarified. Fig. 7 has been remade; figure legends have also been fixed.
  • all english language issues have been fixed using "Grammarly". 

We hope that this manuscript is now acceptable for publication.